# Molecular signature to predict quality of life and survival with glioblastoma using Multiview omics model

Rayan Nassani[1,2], Yahya Bokhari[3,4], Bahauddeen M. Alrfaei[2,5]*

1 Center for Computational Biology, Institute of Cancer and Genomic Sciences, University of Birmingham, Birmingham, United Kingdom, 2 King Abdullah International Medical Research Center (KAIMRC), King Saud bin Abdulaziz University for Health Sciences (KSAU-HS), Riyadh, Saudi Arabia, 3 Department of AI and Bioinformatics, King Abdullah International Medical Research Center (KAIMRC), King Saud Bin Abdulaziz University for Health Sciences (KSAU-HS), Riyadh, Saudi Arabia, 4 Department of Health Informatics, College of Public Health and Health Informatics, King Saud Bin Abdulaziz University for Health Sciences (KSAU-HS), Riyadh, Saudi Arabia, 5 College of Medicine, King Saud Bin Abdulaziz University for Health Sciences (KSAU-HS), Riyadh, Saudi Arabia

* alrfaeiba@ngha.med.sa

**Data Availability Statement:** All relevant data are linked and cited within the paper and its Supporting Information files.

**Funding:** King Abdullah International Medical Research Center (KAMRC) sponsored this work

## Abstract

Glioblastoma multiforme (GBM) patients show a variety of signs and symptoms that affect their quality of life (QOL) and self-dependence. Since most existing studies have examined prognostic factors based only on clinical factors, there is a need to consider the value of integrating multi-omics data including gene expression and proteomics with clinical data in identifying significant biomarkers for GBM prognosis. Our research aimed to isolate significant features that differentiate between short-term ($\leq$ 6 months) and long-term ($\geq$ 2 years) GBM survival, and between high Karnofsky performance scores (KPS $\geq$ 80) and low (KPS $\leq$ 60), using the iterative random forest (iRF) algorithm. Using the Cancer Genomic Atlas (TCGA) database, we identified 35 molecular features composed of 19 genes and 16 proteins. Our findings propose molecular signatures for predicting GBM prognosis and will improve clinical decisions, GBM management, and drug development.

## Introduction

GBM is the most malignant form of brain cancer. It is a grade IV astrocytoma, which accounts for 60–70% of all gliomas [1]. GBM has a median survival rate of 12–18 months post-diagnosis, and only 6.8% of patients survive to 5 years [2]. Survival analysis, prognosis, and the patients' stratification, play a significant role in the successful diagnosis, management, and treatment of cancer [3]. In addition, quality of life (QOL] and survival are influential prognostic factors in neurooncological clinical assessment and therapeutic development [4]. A variety of signs and symptoms present in GBM patients have a negative impact on their QOL and level of independence. These symptoms include neurological deficits, personality changes, epileptic seizures, and cognitive problems [5]. In this sense, patients with poor QOL and long survival rates could be considered a heavy social and financial burden, given their requirements for frequent

under protocol No. RC13/258/R and and NRC.21R.093.03. The funders had no role in study design, data collection and analysis, decision to publish, or preparation of the manuscript.

**Competing interests:** All authors report no competing interests.

care, medical interventions, and hospitalizations [5]. Although GBM patients face devastating problems associated with their QOL, the lack of comprehensive studies on GBM prognosis and other related factors has limited the development of proper guidelines for managing and improving QOL in GBM patients [4].

Integrative omics is a promising approach to unlocking new insights into GBM prognosis mechanisms and identifying new biomarkers and therapeutic targets [6]. Integrative omics approaches combine different types of omics data, such as genomics, transcriptomics, proteomics, and metabolomics, to obtain a more comprehensive understanding of biological processes and pathways in complex biological systems, including cancers [6]. Using the integrative omics approach could reveal previously unknown connections and relationships between molecules, pathways, and biological processes in GBM prognosis. In addition, integrating omics data with patient-level clinical data to identify molecular signatures or biomarkers can help predict disease risk, prognosis, and treatment response, which can be particularly valuable in personalized medicine and individualized treatment plans. Several studies have demonstrated the effectiveness of integrative omics and machine learning approaches in improving our understanding of different cancers, including GBM [7–9]. One study showed that integrating gene expression and copy number data improved the accuracy of GBM subtype classification [10]. Another study showed that integrating single nucleotide polymorphism (SNP), DNA copy number, DNA methylation, mRNA expression, and clinical data identified new GBM subtypes with distinct clinical outcomes [11]. Furthermore, integrative omics approaches have also been used to identify potential therapeutic targets for GBM [12, 13]. Despite the growing interest in this approach, integrative omics is still missing in QOL studies. Moreover, evidence suggests that survival length and QOL are positively correlated in patients with advanced cancer stages [14]. In this study, we integrated gene expression and proteomics data with GBM patients' clinical data, including KPS and overall survival (OS), to predict the molecular signatures associated with enhanced GBM patients' survival and QOL. We developed a Multiview model using the iRF algorithm by training each omics dataset separately, then by training the integrated omics dataset to predict molecular signatures that affect KPS and OS. We then biologically interpreted our findings to determine biological pathways that influence QOL and survival. We then validated the predictions of the Multiview iRF model (MiRF) using Kaplan–Meier (KM) and regularized Cox proportional hazards model (CPH). A similar survival workflow analysis was done previously [15, 16]. Our goal was to identify molecular signatures for GBM patients who survived $\geq$ 2 years with a QOL $\geq$ 80 KPS. We intended to differentiate between short-term ($\leq$ 6 months) and long-term ($\geq$ 2 years) GBM survival, as well as between high KPS $\geq$ 80, and low KPS $\leq$ 60. KPS system represents three main categories: KPS $\leq$ 80, which represents the ability to work and carry on normally with no special care needed; KPS $\leq$ 60, which requires occasional assistance and impairs work but allows self-care; and KPS $\leq$ 40, which renders patients unable to look after themselves, necessitates hospital care, and the disease progresses rapidly. This means QOL $\geq$ 80 KPS represents a quality of life equal to or better than KPS group 80, which represents the ability to work and carry on normal tasks with no special care needed. We propose potential molecular signatures and therapeutic targets that are promising and novel for GBM management and treatment.

## Methods

### Data collection and pre-processing

The GBM data were obtained from the Cancer Genomic Atlas (TCGA) database (https://www.cancer.gov/tcga). The mRNA expression and proteomics datasets were downloaded from the Broad Genome Data Analysis Center (GDAC) Firehose (https://gdac.broadinstitute.org/

accessed on 1 June 2022) as log-transformed, z-score standardized Affymetrix U133 microarray data and z-score standardized RPPA protein expression measured by reverse-phase protein array, respectively. Clinical data were obtained from the cBioportal database (https://www.cbioportal.org/ accessed on 1 June 2022). To download all data as a tar.gz file, use (https://cbioportal-datahub.s3.amazonaws.com/gbm_tcga.tar.gz/ accessed on 1 June 2022).

Data processing was performed using R Statistical Software (R Core Team, 2021), and RStudio (Rstudio Team, 2022.02.3). A general data processing workflow was applied. In the cleaning step, all variables with at least 80% missing data were excluded. Then, in the imputation step, the K-nearest mean approach was applied to replace the missing values using default parameters in the "impute" R package.

For validation log-transformed, z-score standardized mRNA Seq FPKM data and Z-scores of protein abundance ratio measured by mass spectrometry GBM data generated by the National Cancer Institute Clinical Proteomic Tumor Analysis Consortium (CPTAC) were obtained from the cBioportal database.

## Differentially expressed genes (DEGs) screening

We extracted samples with OS $\leq$ 6 months and OS $\geq$ 2 years from the total sample pool of 596. This resulted in 241 samples for gene expression data and 125 samples for proteomics data as our final datasets. Then, to decrease the dimensionality of the gene expression data, we screened the differentially expressed genes (DEGs) between 140 patients who survived for less than 6 months and 101 patients who survived for more than 2 years using the "limma" R package (Ritchie, M.E. et al, 2015) [17]. The cut-off criteria for determining DEGs were false discovery rate (FDR) < 0.05 and absolute log2-FC > 2, 1, and 0.5. For the proteomic data, relevant proteins were previously determined from the TCGA database. We transformed OS months and KPS into binary data.

## MiRF model

In this study, we used the "iRF" R package (Sumanta Basu and Karl Kumbier, 2018) [18] (Fig 1). First, the iRF model grows the reweighted RF with K iterations. Then, significant features that had the highest mean decrease in Gini impurity were stored. Second, iRF applies generalized RIT to projected binary features from reweighted RF to recover important interactions between those features. Third, iRF aggregates interactions prevalent in B bootstrapped samples to evaluate their stability.

Three steps of bootstrap resampling were applied to the MiRF model: the inner layer bootstrapping, which bootstrap samples from input data to build up each tree when growing weighted RF; outer layer bootstrapping, which bootstrap samples from the training data used in the last iteration of iRF to assess the stability of the recovered interactions; and bootstrap resampling, which bootstrap samples from the whole dataset to do R iterations of the model through these samples to recover important features that persisted in at least 50% of the bootstrap replicates and all interactions between those features with a stability score > 0.5.

## Model construction and training

We trained each omics dataset separately, then trained the integrated form of them. First, the gene expression dataset was trained concerning OS to find significant molecular signatures for people who lived $\geq$ 2 years. Then, we trained the model again, but concerning KPS, to find significant molecular signatures for people with KPS $\geq$ 80. We repeated the same process for proteomics and integrated omics datasets. We had 6 models, 3 for OS and 3 for KPS. We partitioned all datasets into 80% training and 20% testing.

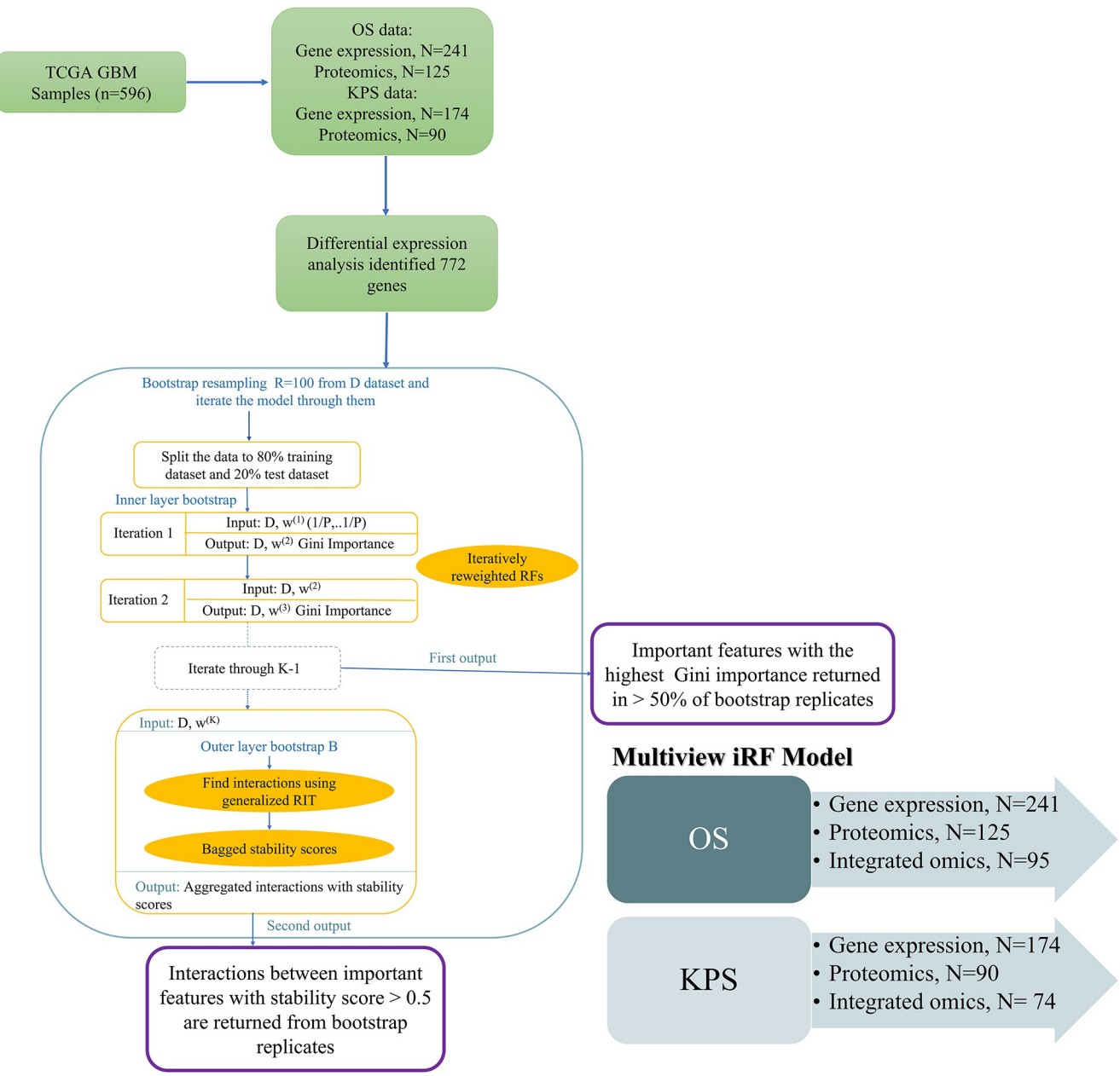

**Fig 1. Multiview iterative random forests (MiRF) model.** The iRF algorithm has 3 main components: first, iterative reweighted RF with K times of iterations; second, generalized RIT that takes projected binary features from the last feature-weighted RF as input; and third, bagged stability scores that aggregate interactions that are prevalent in B bootstrapped samples. The 3 steps of bootstrapping indicated in blue. The model had 2 main outputs: important features with the highest Gini importance returned in at least 50% of bootstrapped replicates, and interactions between important features with stability scores > 0.5 were returned from bootstrapped replicates. The Multiview IRF model is composed of six models: Three models for OS include gene expression, proteomics, and integrated omics; and three for KPS include gene expression, proteomics, and integrated omics.

## iRF tuning parameters

We used the default parameters of the "iRF" R-package to train the MiRF model. In addition to the RF and RIT parameters, iRF had 2 additional parameters: the number of bootstrapped samples (B) and the number of iterations (K). Since the larger values of B improved the

certainty of aggregated interactions, we set $B = 50$, trading off computational cost. We reported the final interactions with K determined by 10-fold cross-validation. For all models, we set $K = 5$. We added one additional parameter which is bootstrap resampling replicates (R); this parameter represents the number of runs we have done for each model using bootstrapped samples. We set $R = 100$.

## iRF performance evaluation and feature selection

The iRF applies the stability principle to recover important interactions. This principle used a consistent set of features along decision paths and bagged stability scores to recover interactions with high consistency throughout the RF [18]. In addition to the stability principle, bootstrap resampling (R) was applied to iterate the model 100 times throughout this dataset.

To assess the predictive accuracy of the iRF model, the area under the precision curve (AUPR) was calculated for each bootstrap replicate using the "PRROC" R package (Jens Keilwagen et al, 2014) [19], and the confidence interval and p-value for all replicates were calculated. Then, the quality of the interaction stability score was evaluated by calculating the area under the receiver operating characteristic curve (AUROC) for each bootstrap replicate using the "AUC" R package (Michel Ballings and Dirk Van den Poel, 2022) [20], and the confidence interval and p-value for all replicates were calculated. Additionally, according to Basu et al., we considered interactions between active features only as true positives and interactions with non-active features as false positives.

As a final output from the last iteration of reweighted RF, we retained features with a mean decrease in Gini impurity $\geq 1$ and persisted in at least 50% of bootstrap replicates for each omics dataset. In addition, we recovered the interactions that were between active features only and had a high stability score $> 0.5$ in the bootstrap replicates for each omics dataset.

## Functional annotation and enrichment analysis

The DAVID [21, 22] database was used to functionally annotate the genes and proteins we obtained from the MiRF model. We used three modules in our analysis: gene ontology (GO), interactions, and pathways. The first step in functional annotation was connecting genes and proteins with their GO terms. We extracted the GO terms of biological process (BP) and molecular function (MF) by choosing the GO direct category. Then, the interactions module was used to recover BioGRID interactions of MiRF-predicted features with GBM driver genes. Finally, the pathways module was used to extract the KEGG pathways associated with our features. We used the STRING [23–25] database to extract protein–protein interactions and compare them with our recovered interactions.

## Survival analysis and Cox proportional hazards model (CPH)

The Kaplan–Meier (KM) method was applied to estimate the survival function of the MiRF-predicted features on the same omics data from the TCGA database. We categorized these features in the omics data as high expression and low expression, using their mean expression as our cutoffs. Then, we applied the log-rank test to compare the survival curves of the 2 categories for each feature. Chi-squared distribution was used to derive a p-value, and a result with $p < 0.05$ was considered significant.

The regularized CPH regression model with the elastic net penalty was applied to the MiRF-predicted features to determine the contribution of each feature to GBM OS prediction. "glmnet" R package (Noah Simon et al, 2011) [26] was used to fit the regularized CPH regression model. We ran a 10-fold cross-validation over the path of λ values obtained from fitting the regularized CPH regression model to find the optimal value of λ. Finally, we chose the λ

value that maximized the Harrell C-index value and applied it to our final regularized CPH regression model. Features with non-zero β value are considered important for OS prediction.

Finally, the univariate and multivariate CPH was used to study the systematic effects of all features on OS. A backward stepwise approach was applied to determine the effect of non-significant features (features with zero β value) on the OS. The concordance index (c-index) was calculated to evaluate the CPH model prediction accuracy. The likelihood ratio test was used to calculate the p-value of the model. Additionally, the confidence interval and p-value for the reach feature were calculated. KM and CPH were performed using the "survival" (Therneau TM et al, 2000) [27] and "Kassambara, Alboukadel, Marcin Kosinski, Przemyslaw Biecek, and Scheipl Fabian. 2021. «survminer: Drawing Survival Curves using "ggplot2"». https://CRAN.R-project.org/package=survminer.

## Visualization and validation

To visualize the effect of the MiRF-predicted molecular signatures on classification between OS ≤ 6 months and OS ≥ 2 years and KPS ≥ 80 and KPS ≤ 60 groups, we calculated PCA components of the predicted signatures using the entire specific data we extracted from TCGA. Then, we calculated their explained variances to determine how much variance those components can explain in PCA.

Then, to visualize how the final OS and overall signatures can classify between OS ≤ 6 months and OS ≥ 2 years, we applied the t-SNE algorithm, which is a nonlinear dimensionality reduction technique. Perplexity was determined according to this equation: perplexity = $\sqrt{N}$, where N is the number of samples. The other parameters were kept as package defaults. For t-SNE plot generation, we used the "Rtsne" R package (Jesse H. Krijthe, 2015) [28].

To validate the effect of OS and overall signatures on GBM patient stratification, we applied these signatures to CPTAC GBM data. First, we applied the whole OS signatures except for NRG1 which was missed in CPTAC data, and the overall signatures on TCGA data to calculate the expression mean cutoffs for each signature comparing patients who survived ≥ 2 years with patients who survived ≤ 6 months using Bonferroni corrected non-paired t-test. Then, we applied OS and overall signatures on CPTAC data using expression means calculated from TCGA data as expression cutoffs. Finally, we calculated the percentage of GBM patients located in OS ≤ 6 months and OS ≥ 2 years zones.

## Results

### Feature selection and interaction recovery

We used the iRF approach to predict important features that classify patients who lived ≥ 2 years from those who lived ≤ 6 months. In our analyses of gene expression, proteomics, and integrated data, we acquired an area under the precision-recall (AUPR) curve of 0.603 with a 95% confidence interval (CI: 0.6–0.61), 0.638 with a 95% CI (0.635–0.64), and 0.644 with a 95% CI (0.636–0.652), respectively, for K = 5 and R = 100 (Fig 2A and a1-c1 in S1 Fig). These results show that our models have relatively acceptable performance, as an AUC value closer to 1 indicates nonrandomized prediction. Additionally, we used iRF to recover important interactions between these features. We acquired an area under the receiver operating characteristic (AUROC) curve of gene expression, proteomics, and integrated data of 74.95% with a 95% CI (73.23%– 76.67%), 77.72% with a 95% CI (75.56%– 79.87%), and 88.78% with a 95% CI (86.25%– 91.3%), respectively, for K = 5 and R = 100 (Fig 2A and a1-c1 in S1 Fig). These results show that our models have reliable performance.

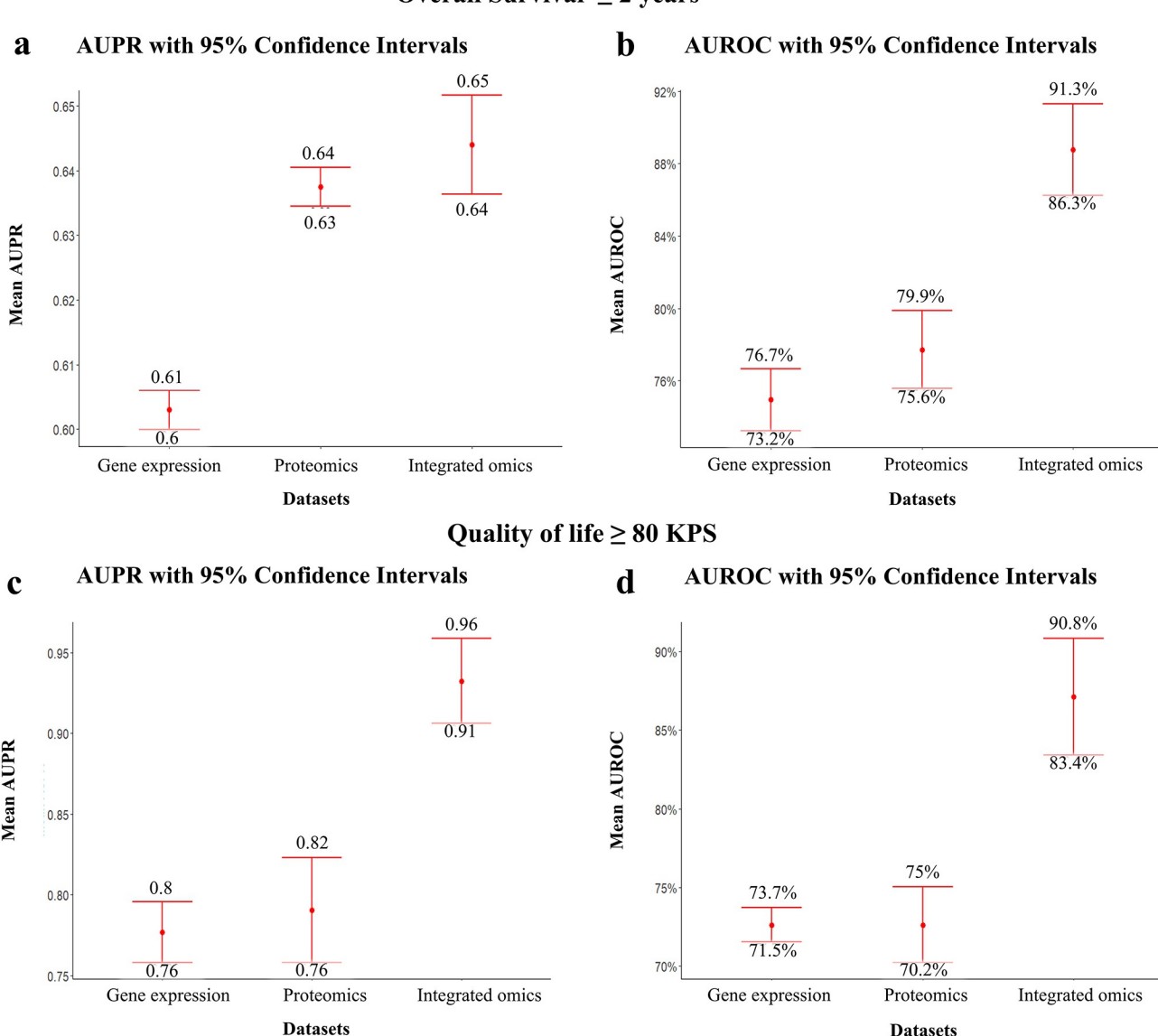

**Fig 2. MiRF model performance in feature prediction and interaction recovery. a** 95% CIs of mean AUPR curves for gene expression, proteomics, and integrated omics data used to predict important features in patients who lived for more than 2 years. **b** 95% CIs of mean AUROC curves for gene expression, proteomics, and integrated omics data used to recover interactions in patients who lived for more than 2 years. **c** 95% CIs of mean AUPR curves for gene expression, proteomics, and integrated omics data used to predict important features in patients who had KPS ≥ 80. **d** 95% CIs of mean AUROC curves for gene expression, proteomics, and integrated omics data used to recover interactions in patients who had KPS ≥ 80.

In the second prediction problem, which focused on classifying patients who had a KPS ≥ 80 from those who had a KPS ≤ 60, we acquired an AUPR curve of gene expression, proteomics, and integrated data of 0.78 with a 95% CI (0.76–0.80), 0.79 with a 95% CI (0.76–0.82), and 0.93 with a 95% CI (0.91–0.96), respectively, for K = 5 and R = 100 (Fig 2B and a2-c2 in S1 Fig). These results show that our models have reliable performance. Then, to recover the important interactions between these features, we obtained an AUROC curve of gene expression, proteomics, and integrated data of 72.63% with a 95% CI (71.55%– 73.7%), 72.63% with a

**Table 1. List of important features obtained by the MiRF model.**

| OS ≥ 2 YEARS | QOL ≥ 80 KPS |
|---|---|
| MRNA DATA | |
| FAM172A/ C5ORF21 | SERPINB10 |
| ZKSCAN3 | TM4SF20 |
| FKBP6 | POU2F3 |
| TRIM62/ DEAR1 | EIF2B5 |
| REST | MLN |
| NOL3 | WRNIP1 |
| CRELD1 | GALK1 |
| DRG2 | RNF121 |
| TNIP1 | RNF6 |
| B3GAT3/ GLCATI | |
| AGFG2/ HRBL | |
| NCKIPSD/ SPIN90 | |
| PROTEOMICS DATA | |
| GAPDH | TFRC |
| TFRC | TUBA1B/Acetyl a Tubulin Lys40 |
| NRG1/HEREGULIN | ERBB2/HER2 |
| PXN/PAXILLIN | RPS6.S6_pS240_S244 |
| G6PD | PECAM1/CD31 |
| BCL2 | LCK |
| ERRFI1/MIG.6 | NDRG1/NDRG1_pT346 |
| RAF1/C-RAF | CDH1/E-Cadherin |
| FOXO3/FOXO3A/P300 | FOXO3/FOXO3a_pS318_S321/P300 |
| EEF2K | RB1/Rb_pS807_S811 |

95% CI (70%–75%), and 87% with a 95% CI (83%–91%), respectively, for K = 5 and R = 100 (Fig 2B and a2-c2 in S1 Fig). These results show that our models have reliable performance.

The significant features we obtained from the MiRF model are presented in Table 1. It shows 2 lists of features that appeared in 50% of bootstrap replicates with a mean decrease in Gini scores ≥ 1 for both OS ≥ 2 years and QOL ≥ 80 KPS. Gene name annotation and ENTREZ-ID for each feature are listed in S1 Table in S1 File. For OS ≥ 2 years, we obtained 12 genes and 10 proteins. FAM172A, ZKSCAN3, FKBP6, TRIM62, REST, NOL3, CRELD1, and DRG2 were from the gene expression dataset; EEF2K, FOXO3, G6PD, RAF1, GAPDH, TFRC, BCL2, NRG1, PXN, and ERRFI1 from the proteomics dataset; and NOL3, TNIP1, B3GAT3, AGFG2, NCKIPSD, EEF2K, PXN, and NRG1 from the integrated omics dataset. We noticed that NOL3, NRG1, PXN, and EEF2K were persisted outputs when training omics data separately or integrated. In addition, for QOL ≥ 80 KPS, we obtained a final output of 9 genes and 10 proteins. SERPINB10, TM4SF20, POU2F3, EIF2B5, MLN, and WRNIP1 were from the gene expression dataset; RPS6, PECAM1, LCK, CDH1, FOXO3-P, NDRG1, ERBB2, and RB1 from the proteomics dataset; and GALK1, RNF121, RNF6, TFRC, NDRG1, ERBB2, and TUBA1B from the integrated omics dataset. We noticed that NDRG1 and ERBB2 were persistent outputs either when trained omics data separately or as integrated. We also found that TFRC was output from both OS ≥ 2 years and QOL ≥ 80 KPS models, as well as FOXO3. However, it was in phosphorated form in QOL ≥ 80 KPS model.

Recovered interactions are shown in Fig 3, which include stability scores for both OS ≥ 2 years (Fig 3A) and QOL ≥ 80 KPS (Fig 3B). Blue indicates interactions between proteomics

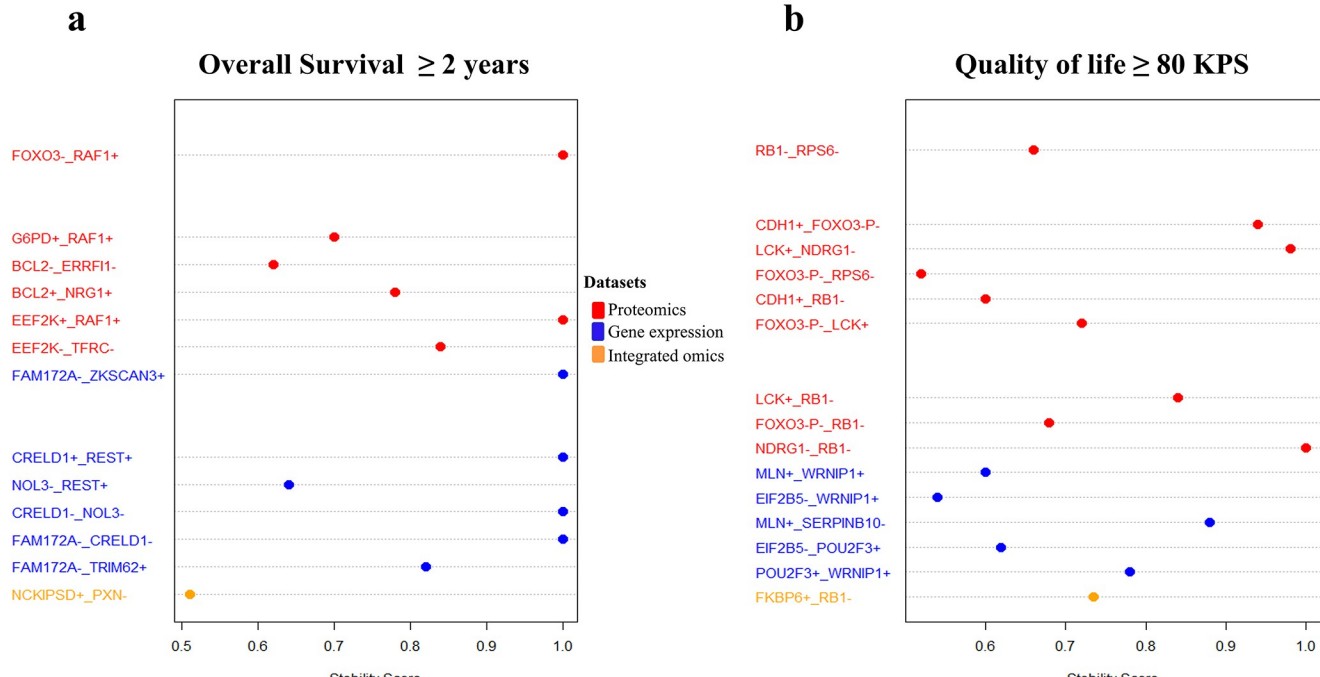

**Fig 3. Stability scores of recovered interactions from the MiRF model. a** Stability scores of recovered interactions for OS ≥ 2 years. **b** Stability scores of recovered interactions for QOL ≥ 80 KPS. Colors indicate interactions from different datasets. Blue indicates interactions between proteomics data, orange indicates interactions between gene expression data, and red indicates interactions between integrated omics.

data, orange indicates interactions between gene expression data, and red indicates interactions between integrated omics. The negative sign represents the inactivation of the target gene, and the positive sign represents its activation. For example, FOXO⁻_ RAF1⁺ means that when FOXO⁻ is inactivated, RAF1⁺ is activated.

## Functional annotation and interaction analysis

Functional annotation and enrichment analysis were done using the DAVID database. We collected GO annotations including BP and MF, BioGRID interactions, and Kyoto Encyclopedia of Genes and Genomes (KEGG) pathways for both lists of genes. (Fig 4 and S2a and S2b Tables in S1 File).

The clustered heatmaps in (Fig 4A) depict the functional annotation obtained from the DAVID database grouped by genes of interest for both OS ≥ 2 years and QOL ≥ 80 KPS. Each cell reports the fold enrichment of each gene toward a specific annotation, with larger values associated with darker coloring and vice versa. Each cluster in the heat map is associated with a specific annotation: the red cluster represents the KEGG pathways, and the blue cluster represents the GO-BP annotation. From the 2 heatmaps, we noticed that some genes from the 2 gene lists shared the same functions and pathways. For GO-BP annotation, we found that NDRG1, BCL2, and FOXO3 play a role in the cellular response to hypoxia; EIF2B5, TFRC, and FOXO3 in aging; SERPINB10, BCL2, NOL3, TFRC, RAF1, EEF2K, and RPS6 in the negative regulation of the apoptotic process; EIF2B5, FOXO3, REST, and RPS6 in the positive regulation of the apoptotic process; BCL2, ERBB2, and NRG1 in positive regulation of cell growth; FKBP6, RAF1, LCK, NRG1, and RB1 in cell differentiation; BCL2, CDH1, and FOXO3 in negative regulation of cell migration; and BCL2, PECAM1, and CDH1 in cell-cell adhesion; BCL2,

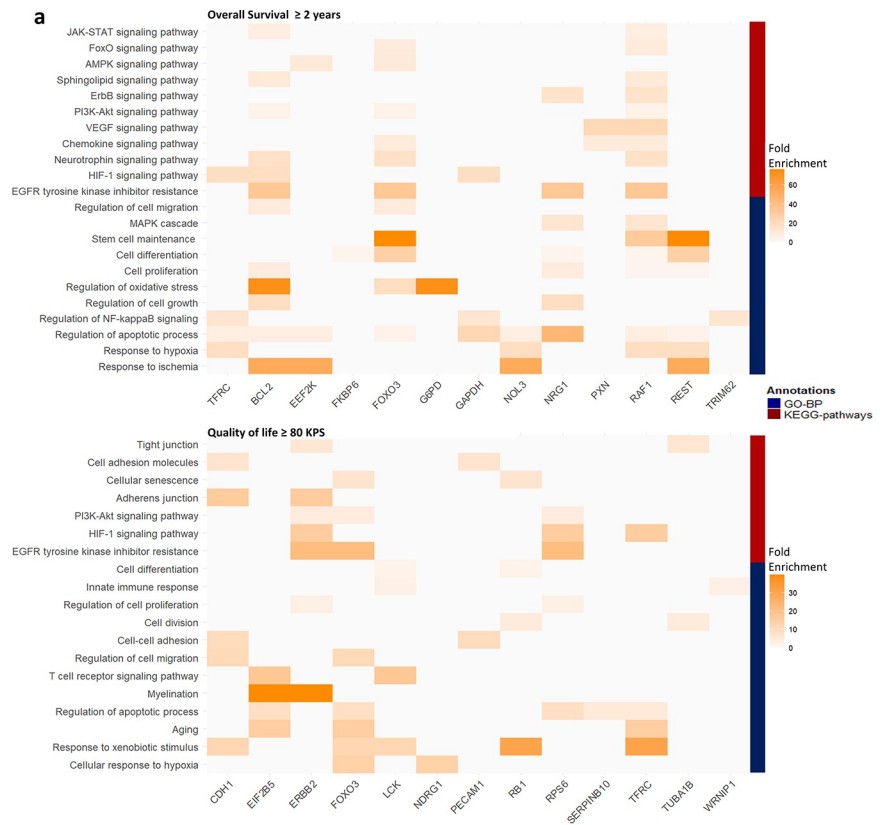

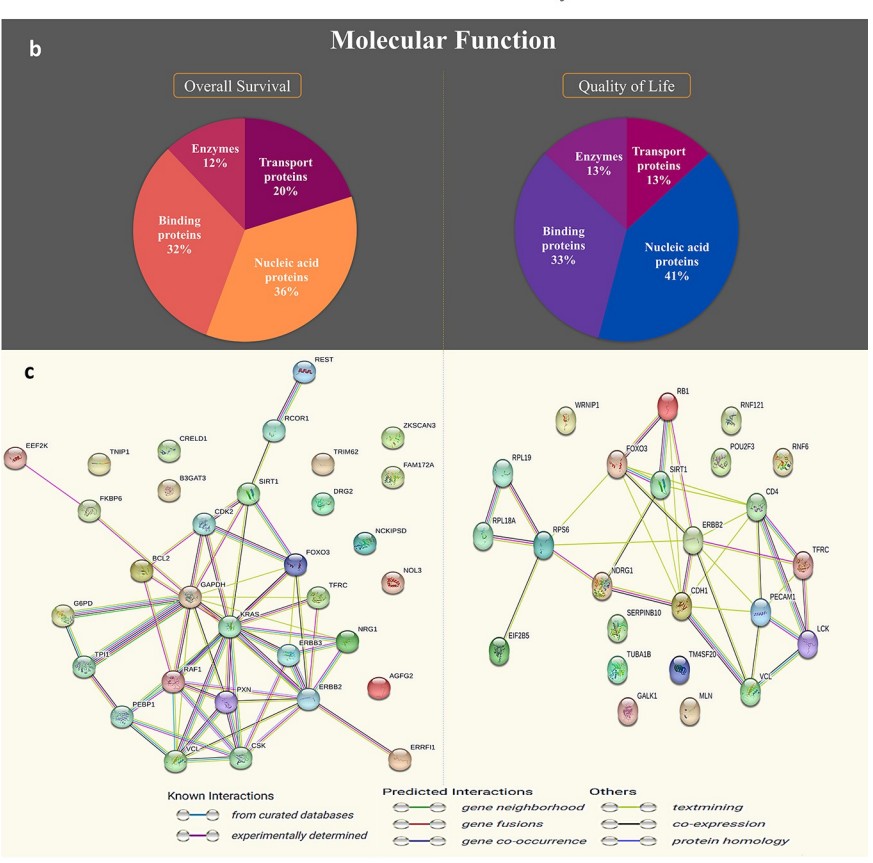

**Fig 4. Functional annotation and interaction analysis. a** Clustered heatmaps illustrate the functional annotation obtained from the DAVID database grouped by the list molecular features obtained from MiRF for OS ≥ 2 years and QOL ≥ 80 KPS. Each cell reports the fold enrichment of each gene toward a specific annotation. Darker coloring is associated with larger values and vice versa. The coloring of clusters corresponds to the following annotations: red represents KEGG pathways and blue represents GO-BP annotations. **b** Pie chart represent distribution of proteins categorized based on functions relevant to overall survival or quality of life genes. **c** Protein–protein interactions from the STRING database for both OS ≥ 2 years and QOL ≥ 80 KPS, respectively. Colored nodes indicate query proteins and the first shell of interactors. Interactions include known interactions from curated databases and those that were experimentally determined; predicted interactions by gene neighborhood, gene fusion, and gene co-occurrence; and others by text mining, co-expression, and protein homology. All the different interactions can be recognized by the different colors. Colored nodes represent query proteins and the first shell of interactors; filled nodes are the 3D structure of known or predicted proteins.

ERBB2, NRG1, and RPS6 in positive regulation of cell proliferation. For the KEGG pathways, we found that BCL2, RAF1, ERBB2, FOXO3, NRG1, and RPS6 have a shared role in EGFR tyrosine kinase inhibitor resistance; BCL2, TFRC, GAPDH, ERBB2, and RPS6 in HIF-1 signaling pathway; BCL2, RAF1, ERBB2, FOXO3, and RPS6 in the PI3K-Akt signaling pathway; and BCL2, PXN, RAF1, and ERBB2 in Focal adhesion; RAF1, ERBB2, and NRG1 in ErbB signaling pathway; BCL2, RAF1, CDH1, ERBB2, and RB1 in pathways in cancer; RAF1 and RB1 in glioma; RAF1 and RPS6 in mTOR signaling pathway; BCL2, RAF1, and TUBA1B in pathways of neurodegeneration; RAF1 and LCK in T cell receptor signaling pathway. S2A and S2B Tables in S1 File show all GO-BP and KEGG pathways including known and novel pathways with references for known pathways. We identified a number of pathways from viral and bacterial infections, as well as other cancer pathways that could be related to GBM survival and QOL.

For GO-MF annotation in (Fig 4B), we found that for OS 36% of its molecular signatures were nucleic acid proteins, 32% were binding proteins, 20% were transport proteins, and 12% were enzymes. For QOL, 41% of its molecular signatures were nucleic acid proteins, 33% were binding proteins, 13% were transport proteins, and 13% were enzymes.

Based on the STRING database results, interaction patterns between important molecular features are shown in Fig 3 for the 2 gene lists of OS ≥ 2 years and QOL ≥ 80 KPS (Fig 4C). These interaction patterns were determined by lab experiments, bioinformatics predictions, co-expression anticipation, and indirect associations. Accordingly, 18% of our recovered interactions with stability score > 0.5 were known direct interactions in the STRING database including FOXO3⁻_RAF1⁺, FOXO3-P⁻_RB1⁻, CDH1⁺_RB1⁻, FOXO3-P⁻_RPS6⁻, and CDH1⁺_FOXO3-P⁻; 43% were known indirect interactions including EEF2K⁻_TFRC⁻, EEF2K⁺_RAF1⁺, BCL2⁺_NRG1⁺, BCL2⁻_ERRFI1⁻, G6PD⁺_RAF1⁺; and 39% were not known interactions. Some existing interactions were experimentally confirmed, and others were computationally predicted.

## Survival analysis and Cox proportional hazards model (CPH)

Survival analysis using KM method and log-rank test (Table 2 and S2 Fig) showed that high expression of FKBP6, REST, EEF2K, MLN, POU2F3, WRNIP1, TM4SF20, and CDH1 was significantly associated with longer survival, with p-values ranging between 0.037 and < 0.001. While low expression of B3GAT3, FAM172A, CRELD1, DRG2, AGFG2, TNIP1, NOL3, RNF6, ERBB2, FOXO3-P, and NDRG1 was significantly associated with longer survival, with p-values ranging between 0.023 and <0.0001. Survival was not affected by the rest of the features. In addition, patients with a high KPS score ≥ 80 showed longer survival rates, with a p-value of <0.0001. According to TCGA clinical data, we found that 63% of patients who lived for more than 2 years had a KPS score ≥ 80.

**Table 2. a. Important features for OS ≥ 2 years. b.** Important features for QOL≥ 80 KPS.

| SYMBOL | Expression status with good survival | P-value | SYMBOL | Expression status with good survival | P-value |
|---|---|---|---|---|---|
| TNIP1 | Low | 0.004 ** | GALK1 | Low | 0.065 |
| B3GAT3 | Low | <0.0001*** | RNF121 | Low | 0.7 |
| NOL3 | Low | 0.0008 *** | RNF6 | Low | 0.00032 *** |
| AGFG2 | Low | 0.023 * | SERPINB10 | Low | 0.3 |
| NCKIPSD | Low | 0.26 | TM4SF20 | High | 0.0042 ** |
| FAM172A | Low | 0.0028 * | POU2F3 | High | 0.0046 ** |
| ZKSCAN3 | High | 0.88 | EIF2B5 | Low | 0.21 |
| FKBP6 | High | 0.0021 * | MLN | High | 0.0057 ** |
| TRIM62 | High | 0.69 | WRNIP1 | High | 0.00085 *** |
| REST | High | 0.021 * | NDRG1 | Low | 0.00026 *** |
| CRELD1 | Low | 0.00049 *** | ERBB2 | Low | 0.011 * |
| DRG2 | Low | <0.0001 *** | TUBA1B | Low | 0.98 |
| NRG1 | Low | 0.2 | RPS6 | Low | 0.095 |
| PXN | Low | 0.31 | PECAM1 | Low | 0.65 |
| EEF2K | High | 0.037 * | LCK | Low | 0.32 |
| GAPDH | Low | 0.17 | CDH1 | High | 0.0022 ** |
| TFRC | Low | 0.24 | FOXO3-P | Low | 0.011 * |
| G6PD | High | 0.21 | RB1 | High | 0.3 |
| BCL2 | High | 0.1 | | | |
| ERRFI1 | Low | 0.065 | | | |
| RAF1 | High | 0.42 | | | |
| FOXO3 | Low | 0.55 | | | |

We applied the regularized CPH regression model with elastic net penalties to the 22 OS features and the 40 overall OS and QOL signatures obtained from the MiRF model. This was done to determine the importance of those features in predicting the survival of patients with GBM. The optimal lambda value was determined using 10-fold cross-validation for both OS and overall signatures models. For the OS model, $\lambda$ = 0.00611 with Harrell C index = 0.72 and standard error = 0.036; for the overall model, $\lambda$ = 0.01549 with Harrell C index = 0.70 and standard error = 0.034 (Fig 5A1 and 5B1). The OS model identified all 22 signatures as nonzero features, which implicates their significance in OS prediction. (Fig 5A2) Furthermore, the overall signature model returned 31 nonzero features out of 40, indicating their significant role in OS prediction. PXN, FKBP6, CRELD1, TM4SF20, RNF6, LCK, RB1, SERPINB10, and RAF1 were zero features which means they have limited or no significance (Fig 5B2).

In addition to the regularized CPH regression model, the univariate and multivariate CPH analyses in Table 3A and 3B demonstrate how our predicted features act individually and together in predicting OS in GBM patients. In the OS ≥ 2 years model (Table 3A), the univariate analysis revealed that the hazard ratio increased significantly by 1.20- to 1.55-fold with high expression of FAM172A, NOL3, CRELD1, DRG2, TNIP1, B3GAT3, AGFG2, NCKIPSD, TFRC, and ERRFI1. In addition, the hazard ratio decreased significantly by 30% with high expression of FKBP6 and BCL2. While multivariate analysis showed that the hazard ratio increased significantly by 1.59- and 2.34-fold with high expression of B3GAT3 and NRG1, respectively, and decreased significantly by 43% and 30% with high expression of G6PD and FOXO3, respectively. Moreover, the backward stepwise approach showed that all 22 OS features have a significant effect on OS. This is consistent with the regularized CPH regression

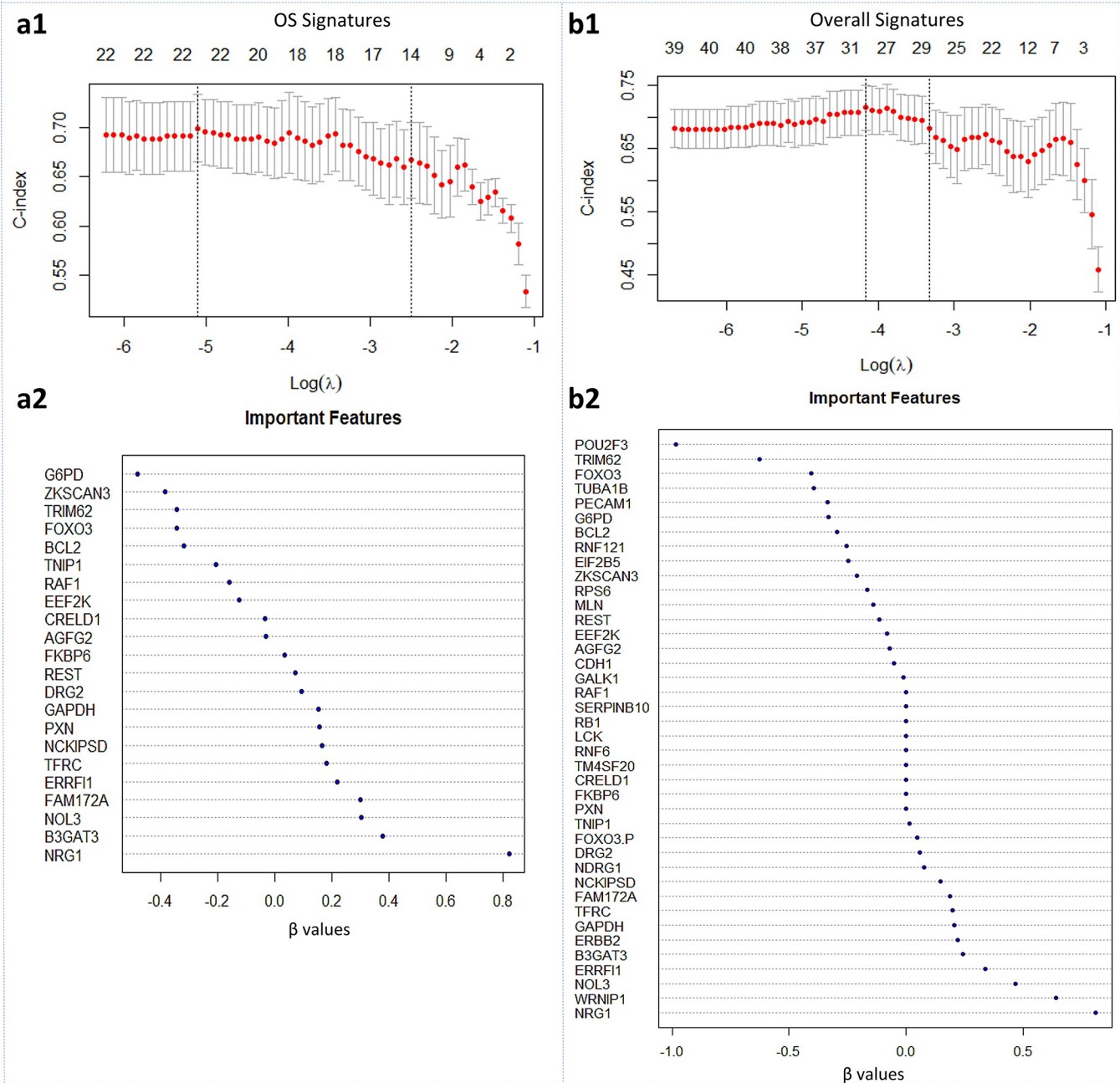

**Fig 5. Regularized CPH regression model. a1** and **b1** Plots of the 10-fold cross-validated error rates show the optimal lambda value with the higher Harrell C-index and minimum error for both OS and overall signatures models, respectively. **a2** and **b2** Dot charts show β values of each feature. Values that are closer to zero are the least important features, and vice versa.

model. The likelihood ratio test p-value for the OS multivariate CPH model was 0.000003, which is significant, and the concordance index was 0.78, which indicates that the multivariate CPH model has nonrandomized performance.

In the overall signature model, we studied the effect of all features from both OS ≥ 2 years and KPS ≥ 80 MiRF models on GBM overall survival (Table 3B). The univariate analysis showed that the hazard ratio increased significantly by 1.55- to 1.63-fold with high expression

**Table 3. a. Univariate and Multivariate Cox Regression Analysis of OS ≥ 2 years. b.** Univariate and Multivariate Cox Regression Analysis of Overall signatures.

| Variables | Univariate Cox Regression Analysis | | | Multivariate Cox Regression Analysis | | |
|---|---|---|---|---|---|---|
| | HR | 95% CI | P-value | HR | 95% CI | P-value |
| **a** | | | | | | |
| **FAM172A** | 1.54 | 1.09–2.17 | 0.0145 * | 1.32 | 0.77–2.27 | 0.31 |
| **ZKSCAN3** | 0.82 | 0.64–1.05 | 0.122 | 0.68 | 0.43–1.05 | 0.08 |
| **FKBP6** | 0.73 | 0.55–0.96 | 0.0257 * | 1.00 | 0.71–1.42 | 0.99 |
| **TRIM62** | 0.87 | 0.67–1.14 | 0.314 | 0.72 | 0.51–1.01 | 0.058 |
| **REST** | 0.82 | 0.67–1.01 | 0.056 | 1.17 | 0.9–1.53 | 0.24 |
| **NOL3** | 1.52 | 1.24–1.87 | 0.00007 *** | 1.48 | 0.98–2.23 | 0.064 |
| **CRELD1** | 1.32 | 1.09–1.6 | 0.0048 ** | 0.94 | 0.65–1.36 | 0.75 |
| **DRG2** | 1.39 | 1.14–1.71 | 0.00148 ** | 1.08 | 0.75–1.56 | 0.67 |
| **TNIP1** | 1.49 | 1.16–1.91 | 0.00179 ** | 0.77 | 0.50–1.18 | 0.23 |
| **B3GAT3** | 1.50 | 1.22–1.85 | 0.00012 *** | 1.59 | 1.00–2.53 | 0.049 * |
| **AGFG2** | 1.27 | 1.05–1.54 | 0.0145 * | 0.90 | 0.61–1.32 | 0.58 |
| **NCKIPSD** | 1.36 | 1.07–1.73 | 0.012 * | 1.06 | 0.66–1.70 | 0.80 |
| **GAPDH** | 1.20 | 0.98–1.47 | 0.083 | 1.18 | 0.81–1.74 | 0.39 |
| **TFRC** | 1.35 | 1.07–1.71 | 0.0113 * | 1.26 | 0.90–1.76 | 0.17 |
| **NRG1** | 1.09 | 0.88–1.36 | 0.433 | 2.34 | 1.32–4.14 | 0.004 ** |
| **PXN** | 1.18 | 0.97–1.43 | 0.099 | 1.18 | 0.86–1.63 | 0.29 |
| **G6PD** | 0.88 | 0.71–1.09 | 0.25 | 0.57 | 0.38–0.84 | 0.005 ** |
| **BCL2** | 0.70 | 0.53–0.92 | 0.0095 ** | 0.73 | 0.51–1.05 | 0.09 |
| **ERRFI1** | 1.27 | 1.03–1.57 | 0.027 * | 1.21 | 0.81–1.79 | 0.35 |
| **RAF1** | 1.00 | 0.83–1.21 | 0.99 | 0.78 | 0.50–1.19 | 0.25 |
| **FOXO3** | 0.90 | 0.75–1.09 | 0.29 | 0.70 | 0.50–0.97 | 0.03 * |
| **EEF2K** | 0.83 | 0.67–1.03 | 0.09 | 0.85 | 0.61–1.2 | 0.35 |
| **b** | | | | | | |
| **FAM172A** | 1.54 | 1.09–2.17 | 0.0145 * | 1.12 | 0.59–2.11 | 0.73 |
| **FKBP6** | 0.73 | 0.55–0.96 | 0.0257 * | 0.88 | 0.51–1.51 | 0.65 |
| **TRIM62** | 0.87 | 0.67–1.14 | 0.314 | 0.42 | 0.26–0.69 | 0.00049 *** |
| **REST** | 0.82 | 0.67–1.01 | 0.056 | 0.87 | 0.64–1.19 | 0.39 |
| **NOL3** | 1.52 | 1.24–1.87 | 0.00007 *** | 1.85 | 1.05–3.25 | 0.032 * |
| **CRELD1** | 1.32 | 1.09–1.6 | 0.0048 ** | 1.07 | 0.73–1.57 | 0.74 |
| **DRG2** | 1.39 | 1.14–1.71 | 0.00148 ** | 1.01 | 0.63–1.63 | 0.95 |
| **TNIP1** | 1.49 | 1.16–1.91 | 0.00179 ** | 1.13 | 0.71–1.79 | 0.61 |
| **B3GAT3** | 1.50 | 1.22–1.85 | 0.00012 *** | 1.32 | 0.79–2.20 | 0.29 |
| **AGFG2** | 1.27 | 1.05–1.54 | 0.0145 * | 0.76 | 0.48–1.18 | 0.22 |
| **NCKIPSD** | 1.36 | 1.07–1.73 | 0.012 * | 1.27 | 0.78–2.08 | 0.34 |
| **GAPDH** | 1.20 | 0.98–1.47 | 0.083 | 1.48 | 0.92–2.38 | 0.10 |
| **TFRC** | 1.35 | 1.07–1.71 | 0.0113 * | 1.37 | 0.9–2.11 | 0.15 |
| **NRG1** | 1.09 | 0.88–1.36 | 0.433 | 2.91 | 1.60–5.28 | 0.0004 *** |
| **PXN** | 1.18 | 0.97–1.43 | 0.099 | 0.85 | 0.56–1.29 | 0.44 |
| **G6PD** | 0.88 | 0.71–1.09 | 0.25 | 0.61 | 0.39–0.97 | 0.036 * |
| **BCL2** | 0.70 | 0.53–0.92 | 0.0095 ** | 0.77 | 0.53–1.14 | 0.19 |
| **ERRFI1** | 1.27 | 1.03–1.57 | 0.027 * | 1.65 | 0.99–2.74 | 0.054 |
| **FOXO3** | 0.90 | 0.75–1.09 | 0.29 | 0.52 | 0.32–0.82 | 0.0053 ** |
| **EEF2K** | 0.83 | 0.67–1.03 | 0.09 | 0.95 | 0.62–1.45 | 0.82 |
| **TM4SF20** | 0.77 | 0.59–0.99 | 0.0428 * | 0.85 | 0.52–1.4 | 0.53 |
| **POU2F3** | 0.73 | 0.5–1.06 | 0.0999 | 0.26 | 0.14–0.46 | 0.000004 *** |

*(Continued)*

**Table 3.** (Continued)

| Variables | Univariate Cox Regression Analysis | | | Multivariate Cox Regression Analysis | | |
|---|---|---|---|---|---|---|
| | HR | 95% CI | P-value | HR | 95% CI | P-value |
| EIF2B5 | 0.89 | 0.73–1.1 | 0.28 | 0.60 | 0.4–0.91 | 0.017 * |
| WRNIP1 | 0.75 | 0.57–0.98 | 0.0326 * | 2.71 | 1.52–4.82 | 0.00069 *** |
| GALK1 | 1.25 | 1.00–1.57 | 0.053 | 0.92 | 0.61–1.38 | 0.69 |
| RNF121 | 0.78 | 0.62–0.99 | 0.039 * | 0.66 | 0.44–0.99 | 0.047 * |
| RNF6 | 1.55 | 1.21–1.98 | 0.000456 *** | 0.93 | 0.64–1.36 | 0.72 |
| TUBA1B | 1.02 | 0.83–1.25 | 0.88 | 0.51 | 0.31–0.85 | 0.00947 ** |
| ERBB2 | 1.37 | 1.14–1.65 | 0.000886 *** | 1.41 | 1.00–1.99 | 0.048159 * |
| RPS6 | 1.16 | 0.93–1.45 | 0.196 | 0.71 | 0.48–1.06 | 0.09 |
| PECAM1 | 0.94 | 0.77–1.16 | 0.57 | 0.60 | 0.31–1.15 | 0.13 |
| NDRG1 | 1.63 | 1.28–2.06 | 0.000056 *** | 0.99 | 0.64–1.55 | 0.97 |
| CDH1 | 0.73 | 0.56–0.95 | 0.018 * | 1.06 | 0.69–1.62 | 0.78 |
| FOXO3-P | 1.51 | 1.15–1.98 | 0.00317 ** | 0.87 | 0.55–1.38 | 0.54 |

of RNF6, ERBB2, NDRG1, and FOXO3-P in addition to the OS features in Table 3A. Additionally, the hazard ratio decreased significantly by approximately 25% with high expression of TM4SF20, WRNIP1, RNF121, and CDH1 in addition to the OS features in Table 3A. While multivariate analysis showed that the hazard ratio increased significantly by 1.41- to 2.91-fold with high expression of NOL3, NRG1, WRNIP1, and ERBB2; and decreased by 70% - 30% with high expression of TRIM62, G6PD, FOXO3, POU2F3, EIF2B5, RNF121, and TUBA1B. Moreover, the backward stepwise approach showed that 34 out of 40 features have an important effect on OS as the model performance was enhanced after removing LCK, RB1, SERPINB10, ZKSCAN3, MLN, and RAF1. These results increased our confidence in the regularized CPH regression model. The likelihood ratio test p-value for the overall signature multivariate CPH model was 0.00000004, which is significant, and the concordance index was 0.81, which indicates that the multivariate CPH model has nonrandomized performance.

## Visualization and clinical application of the molecular signatures

To visualize the effect of MiRF signatures on stratifying GBM patients according to their OS and KPS, we applied PCA algorithm to both OS and QOL signatures. 3D PCA scatter plot illustrates the effect of MiRF-predicted features on separating the different groups. For the OS model, the 22 molecular signatures showed two clear clusters: the blue cluster represents patients who lived $\geq$ 2 years, and the green cluster represents patients who survived $\leq$ 6 months. This plot presents a total of 50% of the explained variance since the first PCA component is 25%, the second component is 15%, and the third component is 10% (Fig 6A1). For the QOL model, the 19 molecular signatures showed two clusters: the blue cluster represents KPS $\geq$ 80, and the green cluster represents KPS $\leq$ 60. This plot presents a total of 45% of the explained variance since the first PCA component is 20% of the variance, the second component is 15%, and the third component is 10% (Fig 6A2).

In the method of visualization, we compared the average expression of all samples for each signature for the different groups of OS and QOL. The heatmap in (Fig 6B1) illustrates the expression mean of OS signatures between OS $\geq$ 2 years and OS $\leq$ 6 months. While the heatmap in (Fig 6B2) compares the expression mean of QOL signatures between KPS $\geq$ 80 and QOL $\leq$ 60. Each cell reports an expression average. Blue shades indicate high expression and

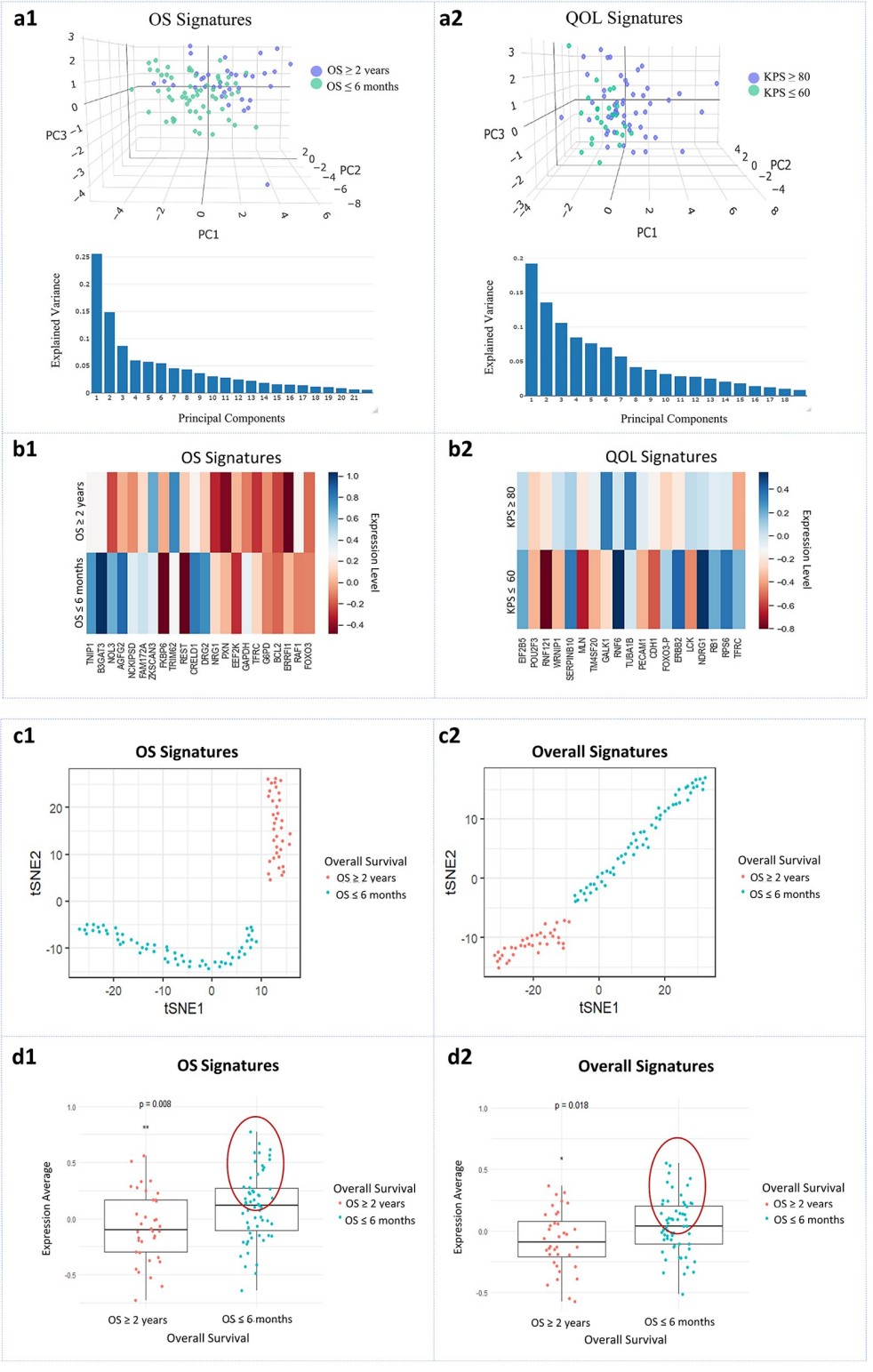

**Fig 6. MiRF signatures validation. a1** and **a2** 3D PCA scatter plots with their explained variance plots for OS and QOL signatures show two clusters for each model: blue for OS ≥ 2 years, green for OS ≤ 6 months and blue for KPS ≥ 80, green for KPS ≤ 60, respectively. **b1** and **b2** Heatmaps represent internal validation on the list of molecular features obtained from MiRF for OS ≥ 2 years and QOL ≥ 80 KPS. Each cell reports the average expression of all samples for each signature. Blue shades indicate high expression and red shades indicate low expression. **c1** and **c2** t-

SNE plots show the effect of OS and overall signatures, respectively, in distinguishing OS ≥ 2 years and OS ≤ 6 months groups. Each plot has two clusters have distinct expression profile of OS and overall signatures. Red for OS ≥ 2 years, green for OS ≤ 6 months. **d1** and **d2** Boxplots represent the expression mean cutoffs for OS ≥ 2 years and OS ≤ 6 months of both OS and overall signatures, respectively. Red for OS ≥ 2 years, green for OS ≤ 6 months. A p-value of < 0.05 was considered significant.

red shades indicate low expression. The two heatmaps show clear expression variation between groups.

To visualize the effect of CPH signatures on classifying GBM OS, we applied t-SNE algorithm to determine if there is a significant separation between OS ≥ 2 years and OS ≤ 6 months. The t-SNE plot for the 22 OS signatures showed two clear clusters: the blue cluster indicates patients who lived ≥ 2 years, and the green cluster reflects patients who survived ≤ 6 months (Fig 6C1). In addition, the t-SNE plot for the 34 overall signatures also showed two clear clusters: the blue cluster represents patients who lived ≥ 2 years, and the green cluster represents patients who survived ≤ 6 months (Fig 6C2).

To validate the effect of CPH signatures on GBM OS stratification and make it clinically applicable, we applied the final set of signatures obtained from CPH models to GBM omics data obtained from CPTAC project. First, we determined expression cutoffs by applying both OS and overall signatures to TCGA data (Fig 6D1 and 6D2). We used the average expression of signatures as our cutoffs. From the boxplots, we notice that patients who survived longer have a low expression average of both OS and overall signatures with a p-value of 0.008 and 0.018, respectively. For OS signatures, applying expression means cutoffs obtained from TCGA data to CPTAC data led to clearly stratifying 40% of the patients to OS ≥ 2 years and OS ≤ 6 months. In addition, for overall signatures, applying expression means cutoffs to CPTAC data led to clearly stratifying 42% of the patients to OS ≥ 2 years and OS ≤ 6 months (Table 4).

We applied previously predicted signatures to TCGA to calculate expression mean cutoffs. We then applied these cutoffs to CPTAC data to see their effect on stratifying GBM patients according to their possible OS. From (Table 4) we can see that our signatures have a 10% - 12% better performance in stratifying GBM patients.

## Discussion

In this study, our goal was to identify the genetic signatures of GBM patients who survived ≥ 2 years with QOL ≥ 80 KPS. We collected gene expression, proteomics, and clinical data from the TCGA database. In differentially expressed genes (DEGs), which have been done to

**Table 4. Expression means cutoffs and external validation on CPTAC database.**

| Signatures | Expression mean for OS ≥ 2 years | Expression mean for OS ≤ 6 months | P-value of comparing the tow means | Percentage of CPTAC samples in mean area |
|---|---|---|---|---|
| OS Signatures | - 0.09 | 0.09 | 0.008 ** | 40% |
| Overall Signatures | - 0.08 | 0.042 | 0.018 * | 42% |
| Yu's Signature [29] | - 0.07 | 0.14 | 0.12 | 32% |
| Yin's Signature [30] | - 0.035 | 0.19 | 0.0097 ** | 32% |
| Pan's Signature [31] | - 0.1 | 0.2 | 0.042 * | 32% |
| Wang's Signature [32] | - 0.095 | 0.27 | 0.00012 *** | 28% |
| Zhang's Signature [33] | - 0.03 | 0.127 | 0.016 * | 30% |
| Cao's Signature [34] | - 0.036 | 0.31 | <0.0001 *** | 31% |

decrease the dimensionality of gene expression data, we used an absolute log2-FC > 2, 1, and 0.5 comparing OS ≥ 2 years with OS ≤ 6 months and KPS ≥ 80 with KPS ≤ 60. FC > 2 did not return any genes, FC > 1 returned only one gene, and FC > 0.5 returned 772 genes for OS comparison and 357 genes for KPS comparison.

We used the iRF algorithm [18] for feature selection in integrative omics data. This method is preferred because it identifies stable and robust high-order interactions between features, maintains high accuracy, and is interpretable. Omics data are often characterized by high dimensionality, missing values, multicollinearity, high noise, normalization problems, imbalance, and complexity [35–38]. Identifying meaningful relationships between variables is critical for understanding complex disease phenotypes and molecular pathways. Compared to other machine learning methods for analyzing integrative omics data, iRF has several advantages. Firstly, it uses a recursive feature elimination approach to iteratively eliminate the least informative features to enhance model accuracy and stability in high-dimensional datasets [18]. Secondly, iRF is designed to identify high-order interactions between features which is essential for integrative omics data where many biological processes involve interactions among multiple molecular features from multiple sources. This makes iRF an excellent method for biological and biomedical applications. Thirdly, iRF has been shown to handle noisy data. Building a new random forest at each iteration eliminates the effect of correlated variables and noisy data on feature importance. The invariance of decision trees also ensures less bias due to the signal-to-noise ratio among biological replicates, a significant concern in omics data analysis. It also deals with imbalanced data without affecting model performance. The iRF algorithm is less prone to overfitting than other machine learning approaches because it utilizes much smaller decision trees, mitigating the problem of overfitting in high-dimensional integrative omics data. The algorithm additionally enhances the interpretability of the RF's recovered interactions by applying the generalized random intersection trees algorithm (RIT) and calculating a stability score [18]. This helps identify stable, robust, and reproducible high-order interactions across different subsets of the same dataset. This is particularly important due to the variability and instability in omics data analysis that arise from the underlying biological processes and environmental factors across multiple samples [18].

The iRF algorithm contains 2 layers of bootstrap resampling [18]. In addition to these 2 layers, we added a third layer of bootstrap resampling to iterate the whole model through the whole samples. The third bootstrap layer ensures that the model is rotated through the entire dataset and improves the reliability of the output by considering the intersection between the collective predictions of bootstrap replicates.

To evaluate the performance of the iRF algorithm, AUPR and AUROC curves were calculated. Score 1.0 indicates that the performance level is perfect and stable, whereas 0.5 indicates that the performance level is random and uncertain. The results of the AUPR curves for all trained datasets were ≥ 0.6, with significant 95% CIs. This means that our predicted features were not randomly selected for all models. In addition, the AUROC curves for all the trained datasets were significant ≥ 70%, with 95% CIs. This indicates that our recovered interactions were stable with high-quality stability scores. We noticed that AUPR and AUROC curves were improved with integrated omics for both the OS ≥ 2 years and QOL ≥ 80 KPS models up to ≥ 0.8, with significant 95% CIs, except for AUPR of OS ≥ 2 years model, which showed no significant improvement.

Since we trained the gene expression and proteomics data separately and concatenated, we obtained information from different points of view from these datasets. For this reason, we called our model a MiRF. Our study complied with the complementary principle by exploring the comprehensive picture of omics datasets. It also complied with the consensus principle through the agreement of some features between the 2 models, which greatly improved our

conclusions. From the OS ≥ 2 years model, we obtained 22 features, including 12 from the gene expression data and 10 from the proteomics data. We noticed that the NOL3 gene and EEF2K protein were persistent outputs from training the omics data separately and as integrated. They were both involved in apoptotic regulation. From the QOL ≥ 80 KPS model, we obtained 19 features: 9 from the gene expression data and 10 from the proteomics data. We noticed that ERBB2, NDRG1, and TFRC proteins were persistent outputs from training the omics data separately and integrated. TFRC and FOXO3 proteins appeared in both the OS ≥ 2 years and QOL ≥ 80 KPS models. The above findings demonstrate the principle of consensus of the MiRF model.

The MiRF identified complex nonlinear interactions between predicted features. It employs a feature importance metric to identify important and significant interactions between features. The feature importance metric assigns a score (mean decrease in Gini impurity) to each feature that measures its contribution to prediction as part of an interaction. The feature importance score is determined by two factors: feature impact which reflects a feature's individual effect on the outcome, and feature importance with respect to interaction which measures the contribution of a feature to complex interactions. This feature importance metric enables iRF to provide more accurate and interpretable predictions while accounting for the complexity of the underlying interactions between features. We also calculated a stability score which is a measure of the robustness of the discovered interactions. The stability score provides an estimate of the probability that a particular interaction is consistently recovered across different bootstrapped samples of the training dataset. We identified 13 important interactions from OS ≥ 2 years models with stability score ≥ 0.5: 6 from proteomics data, 6 from gene expression data, and 1 from integrated omics data. In addition, we identified 15 important interactions from KPS ≥ 80 models with stability score ≥ 0.5: 9 from proteomics data, 5 from gene expression data, and 1 from integrated omics data.

For the proteomics output, we used gene names associated with the protein list in our functional annotations, as gene names are universal. Additionally, we connected the genes' SYMBOL with their ENTREZ-ID identifier, which is a standard and universal identifier throughout large, curated databases. To perform a functional analysis of the set of genes and proteins we obtained from the MiRF model, we used the DAVID database. We connected the gene list with its GO term to annotate them with their biological functions, which made them more understandable and interpretable. We extracted the GO terms of BP and MF, as they are more relevant to cancer prognosis. Based on the functional analysis, we concluded that our predicted genes are involved in important biological functions and pathways that regulate cell growth, proliferation, and cell death. These biological functions and pathways affect cancer prognosis, patients' survival, and treatment outcomes. For example, hypoxia is strongly correlated with the poor prognosis of solid cancers. BCL2 [39], ERBB2 [40], RPS6 [41], RAF1, and FOXO3 [42] participate in the regulation of responses to hypoxia through PI3K/AKT/mTOR, NFB, and MAPK signaling pathways that are also activated in a hypoxia-independent manner by several factors that eventually activate hypoxia-inducible factor (HIF)-1α [43]. Activation of HIF-1α enhances the tumor microenvironment and improves tumor cell survival and propagation through increased blood vessel formation, aggressiveness, metastasis, and resistance to treatment [43]. TFRC [44], PECAM1 [45], GAPDH [46], GALK1 [47], NDRG1 [44], and DRG2 [48] also have role in cellular response to hypoxia. Another important biological function is apoptosis. Lack of apoptotic control allows cancer cells to survive longer, and hence mutations will accumulate, which can increase tumor invasiveness, stimulate angiogenesis, and disrupt cell differentiation [49]. BCL2 [50, 51], ERBB2 [52], RAF1, FOXO3 [53], NRG1 [54], PXN [55], NOL3 [56, 57], TFRC [44], EEF2K [58], EIF2B5 [59], TUBA1B [60], FAM172A [61], RNF6 [62], and RPS6 [63] regulate the apoptotic process through the

epidermal growth factor receptors (EGFR), FOXO, NF-B, JAK-STAT, and PI3K/AKT signaling pathways. GAPDH also has a role in neuron apoptotic processes under oxidative stress [46]. Cellular responses to oxidative stress and the regulation of reactive oxygen species (ROS) are double-edged swords in cancer progression and prognosis. Increasing metabolic rate, mutation of genes, and hypoxia result in ROS production in tumor cells. Moderate levels of ROS are quenched by the increased activation of antioxidant pathways in cancer cells that promote tumor progression. However, programmed cell death, including apoptosis, autophagy, and necrosis, can also be triggered by ROS and hence hinder cancer progression [64]. BCL2 [65], G6PD [66], NOL3 [56], RAF1, and FOXO3 [53] are involved in different signaling pathways that regulate reactive oxygen species' metabolic process. Enhanced cell proliferation correlates strongly with a poor prognosis of cancer [67]. BCL2 [68], ERBB2 [40], NRG1 [54], RPS6 [63], RAF1, FOXO3 [53], REST [69], PXN [55], B3GAT3 [70], RNF6 [62], EEF2K [71], ERRFI1 [72], TUBA1B [60], FAM172A [61], ZKSCAN3 [73], TRIM26 [74], RNF121 [75], and TNIP1 [76] regulate cellular proliferation through the ErbB, EGFR, RAS-RAF-MEK-ERK MAPK, PI3K-AKT-mTOR, and NF-B signaling pathways [77]. Epithelial-to-mesenchymal transition (EMT) plays an essential role in the invasion and metastasis of different cancers, including gliomas. EMT is linked to cancer stem cell properties and chemotherapeutic resistance [78]. ZKSCAN3 [73], TRIM26 [74], FAM172A [61], B3GAT3 [70], FOXO3 [79], CDH1 [78], and REST [80] regulate EMT through FAK/AKT, Ras/ERK MAPK-PI3K/Akt, TGF/ P38MAP signaling pathways. It is worth noting that all mentioned genes affect CDH1 expression which is controlled by SMARCA5, a stem cells differentiation gene and cancer marker [81]. In addition to the well-known biological functions and pathways, we identified a novel association of neurodegenerative, cardiac, vascular, viral, bacterial, and other cancer pathways with GBM (S2A and S2B Tables in S1 File). These findings require further investigation, and they could implicate common therapies between those diseases and GBM depending on the gene expression and proteomics profiles of the patient which improves personalized medicine. Our findings come along with Alzahrani et al findings who found pathways associated with COVID-19 poor prognosis in GBM patients [82]. BIOGRID interactions analysis showed that our predicted molecular features interact with GBM driver genes including PIK3R1, EGFR, TP53, RB1, and NF1 (S3D Fig). We also investigated the mutations in GBM driver genes in TCGA data and found that most of the mutations are in GBM patients with low survival but nonsignificant p-values (S3A–S3C Fig).

The STRING database consists of direct (physical) and indirect (functional) protein–protein interactions arising from computation prediction, knowledge transfer between organisms, and interactions gleaned from other primary databases [24]. From our predicted interactions, we found that $CDH1^+\_FOXO3-P^-$ is a known critical functional interaction for cancer progression. Research by Chen et al. has shown that FOXO3 functions as a transcription factor that regulates CDH1 gene expression and thus inhibits the metastatic potential of cancer cells [83]. In addition, $FOXO3^-\ RAF1^+$ is a crucial functional interaction involved in cellular proliferation signaling pathways. In response to external stimuli and growth factors, FOXO3 is negatively regulated by the PI3K-Akt/protein kinase B (PKB) and Ras-RAF-MEK-ERK signaling pathways. Specifically, external stimuli activate Ras, which activates subsequent kinases RAF, mitogen-activated protein kinase kinase (MEK), and extracellular signal-regulated kinase (ERK/MAPK) to phosphorylate FOXO3 [51]. Further investigations into the recovered interactions are required. This could lead to the discovery of novel underlying mechanisms that affect GBM prognosis and hence new therapeutic targets.

Survival analysis was performed on the predicted features for OS $\geq$ 2 years and for the overall OS and QOL features to validate our findings and establish the molecular signature for stratifying GBM patients. A KM curve was used to visualize the differences in survival between

high and low-expression groups for each feature. A log-rank test was conducted to determine whether there was a significant difference between them. Furthermore, we applied the regularized CPH regression model with elastic net penalties to determine the significance of OS features and overall features in predicting GBM overall survival. Finally, we evaluated how our predicted features act individually and together to predict OS in GBM patients using univariate and multivariate CPH analyses. We evaluated the predictive accuracy of the CPH model using the concordance index (c-index), which is the generalization of AUROC for regression problems [84]. The c-index and p-value for both CPH models were significant. From the survival analysis, we found that low expression of NOL3, DRG2, B3GAT3, FAM172A, CRELD1, AGFG2, TNIP1, RNF6, ERBB2, FOXO-P, and NDRG1 enhanced GBM patients' survival and significantly decreased the hazard ratio. Furthermore, high expression of FKBP6, WRNIP1, POU2F3, REST, TM4SF20, EEF2K, and CDH1 enhanced GBM patients' survival and significantly decreased the hazard ratio. However, the hazard ratio for WRNIP1 increased significantly in the multivariate CPH model. From univariate analysis, high expressions of NCKIPSD, TFRC, BCL2, RNF121, and ERRFI1 demonstrated significant effects on the hazard ratio. While from multivariate analysis TRIM62, NRG1, G6PD, FOXO3, EIF2B5, RNF121, and TUBA1B showed significant effects on the hazard ratio. Even though GAPDH, PXN, GALK1, RPS6, RAF1, and ZKSCAN3 showed no statistical significance on survival and hazard ratio, the backward stepwise approach showed that they have an important effect on the other features. Increasingly, we noticed that KPS scores have a significant correlation with survival. From the TCGA data, we found that 64% of patients who survived 2 years or longer had KPS $\geq$ 80. This suggests that the QOL gene signature is also implicated in OS.

From the literature, we found that high expression of NOL3 [85], DRG2 [86], B3GAT3 [87], CRELD1 [88], TNIP1 [76], ERBB2 [89], FOXO3 [42], REST [80], EEF2K [69], TFRC [44, 90], NRG1 [54], TUBA1B [60], PXN [53], GALK1 [47], RPS6 [91], and RAF1 [92, 93] were shown to be poor prognostic indicators in GBM patients and associated with lower overall survival. The survival outcomes reported by Liang et al. on REST using TCGA GBM patients contrast with our findings; this might be because we specifically focused on people who lived more than 2 years and less than 6 months in our survival analysis. Furthermore, we found that high expressions of NDRG1 [94], CDH1 [78], ERRFI1 [72, 95], G6PD [96], and ZKSCAN3 [29, 32] were significantly correlated with better prognosis of GBM patients and longer overall survival. On the contrary, Kathagen-Buhmann et al., reported that knocking down the G6PD gene in glioblastoma xenograft mice resulted in prolonged survival [66].

In a related vein, GAPDH function in tumor cells is controversial; it is important for cancer cell survival, but under oxidative stress, it induces apoptosis. Lazarev et al. found that enhancing the aggregation of oxidized GAPDH is a promising strategy to overcome GBM resistance to therapeutic tools [46]. In addition, Sun et al. showed that downregulation of NCKIPSD was associated with breast cancer and colon cancer recurrence [97]. Yuan et al. identified POU2F3 as a marker for good prognosis of lung adenocarcinoma (LUAD) patients [98]. Chen et al. identified a positive correlation between FAM172A expression and better prognosis of pancreatic cancer patients [61]. Wang et al found an association between AGFG2 expression and prognosis in colorectal cancer patients [99]. Liu et al. reported high expression of RNF6 as an independent poor prognosis indicator in colorectal cancer [100]. Zhao et al. Zhao et al. [101] revealed that RNF121 overexpression inhibits growth and invasion of human renal carcinoma [102]. Lipponen et al. study revealed a significant improvement in the prognosis of patients with breast cancer expressing abnormally high levels of BCL2 [103]. Chen et al. reported that heterozygous loss of TRIM62 correlates significantly with poor OS in breast cancer patients [104]. Jiao et al. and Palaniappan et al. indicated that high expression of EIF2B5 was associated with poor prognosis of liver cancer patients and worse survival of colorectal patients [105, 106].

From the above review, we noticed that all our predicted features were approved experimentally as survival biomarkers for GBM or other cancers. However, WRNIP1, FKBP6, and TM4SF20 were novel. Nevertheless, the expression status of ERRFI1, REST, EEF2K, NDRG1, and RAF1 in survival analyses from the literature did not match our analysis. This suggests that either there is an unknown underlying mechanism for those features in survival, or there are too few samples in our study to make serious conclusions. Further investigation and validation are thus required to support our signatures.

We visualized the effect of MiRF output on separating GBM patients according to their OS and QOL using PCA approach. MiRF molecular signatures showed an accepted effect on GBM patients' classification. Internal validation further demonstrated clear variation in the expression level of each feature between different GBM groups. We also visualized the effect of CPH output on separating GBM patients according to their OS using t-SNE algorithm. Both OS and overall signatures showed clear separation.

Finally, to make our molecular signatures clinically applicable, we determined average expression cutoffs for each signature using TCGA data. We applied these cutoffs to CPTAC GBM data to stratify GBM patients to OS $\geq$ 2 years and OS $\leq$ 6 months. We found that OS signature stratified 40% of GBM patients and overall signature stratified 42%. We compared our signatures with previously predicted signatures and found that our signatures were the most effective at stratifying GBM patients. These signatures will help enhance the clinical plan and provide intense care for patients predicted with short OS to prolong their OS.

Finally, here we provide evidence that combined measurement of expression levels of OS and overall molecular signatures may be informative on the outcome of GBM patients. We anticipate that further investigation with a larger sample size could confirm our findings providing critical information on human GBM malignancy. This could enable improved clinical decisions and GBM management by setting treatment plans with the most appropriate therapeutic strategy. These prospective findings will also improve GBM prognosis and drug development. On the economic side, increasing the accuracy of therapeutic plans, including interventions, will reduce costs for patients, families, and the ecosystem.

## Supporting information

**S1 Fig. AUROC and AUPR curves.** The PR and ROC curves represent the performance of a binary classification model on bootstrapped test datasets with class imbalance (median curves with their AUCs are shown). **a1 & a2** AUPR and AUROC curves of the gene expression MiRF model for OS $\geq$ 2 years and KPS $\geq$ 80 models, respectively. **b1 & b2** AUPR and AUROC curves of the proteomics MiRF model for OS $\geq$ 2 years and KPS $\geq$ 80 models, respectively. **c1 & c2** AUPR and AUROC curves of the integrated omics MiRF model for OS $\geq$ 2 years and KPS $\geq$ 80 models, respectively. An AUC value closer to 1 indicates better performance. In AUPR curves the color scale on the right side of the plot represents the value of the threshold. This threshold is used to calculate the precision and recall values for each point on the curve. Each shade represents a different threshold value. The fact that we have small data points in the test dataset is reflected in the PR curves which display only two colors (thresholds). This means precision and recall values at different classification thresholds may not be well represented. However, we coped with this problem by bootstrapping and taking the median of model performance, and through using a second parameter AUROC. This provides a comprehensive understanding of the model's statistical performance on binary classification data with class imbalance.
(ZIP)

**S2 Fig. Kaplan–Meier analysis. a** KM plots and log-rank test p-values of important genes obtained from the MiRF model for OS $\geq$ 2 years. **b** KM plots and log-rank test p-values of

important proteins obtained from the MiRF model for OS $\geq$ 2 years. **c** KM plots and log-rank test p-values of important genes obtained from the MiRF model for KPS $\geq$ 80. **d** KM plots and log-rank test p-values of important proteins obtained from the MiRF model for KPS $\geq$ 80. **e** KM plot and log-rank test p-value of KPS. A p-value of < 0.05 was considered significant.
(ZIP)

**S3 Fig. Mutations in the driver genes in our specified TCGA dataset. a** Box plot compares frequencies of driver mutations between high and low survival groups. **b** bar plot compares frequencies of driver mutations in high and low survival groups. **c** The table shows data behind Fig a and b. **d** The table shows the BIOGRID interactions of MiRF predicted genes with the driver genes. A p-value of < 0.05 was considered significant.
(TIF)

**S1 File.**
(DOCX)

## Acknowledgments

We thank our colleagues at Birmingham University and KAIMRC for their support, comments, and critique of the project: Danesh Moradigaravand, Justina Žurauskienė, Mamoon Rashid, and Jean-Baptiste Cazier.

## Author Contributions

**Conceptualization:** Rayan Nassani, Bahauddeen M. Alrfaei.

**Data curation:** Rayan Nassani.

**Formal analysis:** Rayan Nassani.

**Funding acquisition:** Bahauddeen M. Alrfaei.

**Investigation:** Rayan Nassani.

**Methodology:** Rayan Nassani, Yahya Bokhari, Bahauddeen M. Alrfaei.

**Project administration:** Bahauddeen M. Alrfaei.

**Resources:** Bahauddeen M. Alrfaei.

**Software:** Rayan Nassani.

**Supervision:** Bahauddeen M. Alrfaei.

**Validation:** Rayan Nassani.

**Visualization:** Rayan Nassani, Yahya Bokhari, Bahauddeen M. Alrfaei.

**Writing – original draft:** Rayan Nassani.

**Writing – review & editing:** Yahya Bokhari, Bahauddeen M. Alrfaei.

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
