## [Decision Letter · Decision Letter 0]

5 Apr 2023

PONE-D-23-05111Molecular signature to predict quality of life and survival with glioblastoma using Multiview omics modelPLOS ONE

Dear Dr. Alrfaei,

Thank you for submitting your manuscript to PLOS ONE. After careful consideration, we feel that it has merit but does not fully meet PLOS ONE’s publication criteria as it currently stands. Therefore, we invite you to submit a revised version of the manuscript that addresses the points raised during the review process.

We look forward to receiving your revised manuscript.

Kind regards,

Aniruddha Datta

Academic Editor

PLOS ONE

Journal Requirements:

"We thank our colleagues at Birmingham University and KAIMRC for support, comments, and critique to the project: Danesh Moradigaravand, Justina Žurauskienė, Mamoon Rashid, and Jean-Baptiste Cazier. We thank the King Abdullah International Medical Research Center (KAMRC) for sponsoring this work under protocol No. RC13/258/R and NRC.21R.093.03."

"King Abdullah International Medical Research Center (KAMRC) sponsored this work under protocol No. RC13/258/R.

The funders had no role in study design, data collection and analysis, decision to publish, or preparation of the manuscript. "

"All authors report no competing interests."

6. We note that Figure 3b and 3c in your submission contain copyrighted images. All PLOS content is published under the Creative Commons Attribution License (CC BY 4.0), which means that the manuscript, images, and Supporting Information files will be freely available online, and any third party is permitted to access, download, copy, distribute, and use these materials in any way, even commercially, with proper attribution. For more information, see our copyright guidelines: http://journals.plos.org/plosone/s/licenses-and-copyright.

a. You may seek permission from the original copyright holder of Figure 3b and 3c to publish the content specifically under the CC BY 4.0 license. 

7. Please include a copy of Table 2 which you refer to in your text on page 14.

Reviewers' comments:

Reviewer's Responses to Questions

**Comments to the Author**

1. Is the manuscript technically sound, and do the data support the conclusions?

Reviewer #1: Partly

Reviewer #2: Partly

2. Has the statistical analysis been performed appropriately and rigorously? 

Reviewer #1: I Don't Know

Reviewer #2: I Don't Know

3. Have the authors made all data underlying the findings in their manuscript fully available?

Reviewer #1: No

Reviewer #2: Yes

4. Is the manuscript presented in an intelligible fashion and written in standard English?

Reviewer #1: Yes

Reviewer #2: Yes

5. Review Comments to the Author

Reviewer #1: In this study, the authors proposed a bioinformatics analysis to predict quality of life and survival with glioblastoma using multi-omics model. Although the performance seems promising, some major points should be addressed as follows:

1. All analyses were conducted on TCGA public data without any validation.

2. At least mutations in the driver genes should be investigated.

3. It is unclear why iRF has been used as a feature selection method rather than the others. This step even can be done using traditional approaches i.e., univariate, statistical tests, etc. Thus, the authors should describe this part clearly or have a baseline comparison on it.

4. The authors reported ROC AUC and PRC but they did not show ROC curves, PR curves.

5. More references on bioinformatics workflow i.e., PMID: 35851932, PMID: 34572330 should be added to attract a broader readership.

6. Literature review is insufficient; the authors may improve it using some related works on bioinformatics-based glioblastoma.

7. Quality of figures should be improved.

Reviewer #2: The authors have applied the iRF algorithm to differentiate between short and long-term GBM survival and KPS and then computationally validated their findings. The overall material presented in this paper has integrated quality of life metrics into the study of quantitative cancer analysis, which is not often done in the field. Following is the list of questions and suggested areas of improvement for the authors:

1. The materials and methods section of this paper needs to be presented much before the results sections of the paper as it makes it difficult to interpret the results without understanding the mechanics of all the algorithms involved. The description of methods in the results section is breif and vague and leaves the readers with lots of questions regarding the methods. I would suggest the authors to explain their analysis pipeline well in the introduction section, if they are unable to move the methods section prior to the results. Keeping the diverse background of the PLOS ONE readership in mind, it is important to note that not all readers will be aware of methods used in this paper, prior to reading the methods section.

2. The abstract and introduction introduces the topic of integrative omics, the authors need to clarify the meaning of this term very lucidly in these early sections of the paper. For example, they can state that it integrates omics data with patient-level clinical data to identify biomarkers or molecular signatures.

3.The authors to explain the KPS scale in detail in the introduction section. For example what does it mean to say QOL >= 80 KPS.

4. Please explain why iRF method was used, how it is relevant to your dataset, and what other methods did you consider.

5.In the result sections , page 10, paragraph 1: The authors should explicitly mention that they are using iRF to classify between long and short term GBM survival, if not please provide more clarification of the objective here.

6.In the results sections the authors have presented AUPR scores with confidence intervals multiple times, they need to provide some qualitative explanation or significance of the results to the readers here. Also the authors need to clarify the parameters K=5 and Ba =10, and how these values were determined.

7. The authors state that "Additionally, we used iRF to recover the important interactions between these features." Can they explain what is the meaning of the term "important interactions", what is the process involved here in determining these interactions ? Is this a pairwise score between features or is a feature importance metric ?

8.Please explain the meaning of stability scores for example in the line "Recovered interactions are shown in Figure 2, which include stability scores for both OS ≥ 2years (Fig. 2a) and QOL ≥ 80 KPS (Fig. 2b)."

9.In the functional annotation and interaction analysis section, are any of the pathways the author found previously implicated in GBM or other cancers increased survival and positive clinical outcomes , i.e. OS>=2 and QOL >= 80 KPS ? Or are they new to the GBM literature ?

10. The authors need to clarify how the elastic net penalties were calculated or assigned for CPH regression model.

11. Please explain if the PCA validation was done on the entire dataset or on a specific hold out or test data set ? This needs to be clearly explained as validation must be carried out on a dataset on which the models were not trained.

12.Please cite the R Studio, here is a link on how to do it: https://psyteachr.github.io/quant-fun-v2/citing-r-rstudio.html

13.Figure 1 has only labels 'a' and 'b', the other labels mentioned in the figure caption are missing.

14. The overall english language should be reviewed thoroughly, there are typos in the manuscript, the gene names should be italicized, and figure resolutions need improvement.

6. PLOS authors have the option to publish the peer review history of their article (what does this mean?). If published, this will include your full peer review and any attached files.

Reviewer #1: No

Reviewer #2: No

---

## [Author Response · Author response to Decision Letter 0]

20 May 2023

See attachment at the end, Page 168

---

## [Decision Letter · Decision Letter 1]

5 Jun 2023

Molecular signature to predict quality of life and survival with glioblastoma using Multiview omics model

PONE-D-23-05111R1

Dear Dr. Alrfaei,

We’re pleased to inform you that your manuscript has been judged scientifically suitable for publication and will be formally accepted for publication once it meets all outstanding technical requirements.

Kind regards,

Aniruddha Datta

Academic Editor

PLOS ONE

Additional Editor Comments (optional):

Reviewers' comments:

Reviewer's Responses to Questions

**Comments to the Author**

1. If the authors have adequately addressed your comments raised in a previous round of review and you feel that this manuscript is now acceptable for publication, you may indicate that here to bypass the “Comments to the Author” section, enter your conflict of interest statement in the “Confidential to Editor” section, and submit your "Accept" recommendation.

Reviewer #1: All comments have been addressed

Reviewer #2: All comments have been addressed

2. Is the manuscript technically sound, and do the data support the conclusions?

Reviewer #1: Partly

Reviewer #2: Yes

3. Has the statistical analysis been performed appropriately and rigorously? 

Reviewer #1: Yes

Reviewer #2: Yes

4. Have the authors made all data underlying the findings in their manuscript fully available?

Reviewer #1: Yes

Reviewer #2: Yes

5. Is the manuscript presented in an intelligible fashion and written in standard English?

Reviewer #1: Yes

Reviewer #2: Yes

6. Review Comments to the Author

Reviewer #1: My previous comments have been addressed well. Thus, I suggest it for publication on the current form.

Reviewer #2: I commend the authors for making the changes and substantially improving the quality of this article. This will be an important contribution to the area and interesting paper for the scientist in the community to read. Good luck and well done !!

7. PLOS authors have the option to publish the peer review history of their article (what does this mean?). If published, this will include your full peer review and any attached files.

Reviewer #1: No

Reviewer #2: No

---

## [Editor Report · Acceptance letter]

13 Jun 2023

PONE-D-23-05111R1 

Molecular signature to predict quality of life and survival with glioblastoma using Multiview omics model 

Dear Dr. Alrfaei:

I'm pleased to inform you that your manuscript has been deemed suitable for publication in PLOS ONE. Congratulations! Your manuscript is now with our production department. 

Kind regards, 

on behalf of

Dr. Aniruddha Datta 

Academic Editor

PLOS ONE